# Barriers and enablers to blood culture sampling in Indonesia, Thailand and Viet Nam: a Theoretical Domains Framework-based survey

Pornpan Suntornsut,[1] Koe Stella Asadinia,[2] Ralalicia Limato ![ORCID],[2,3] Alice Tamara,[2] Linda W A Rotty,[4] Rendra Bramanti,[5] Dwi U Nusantara,[6] Erni J Nelwan ![ORCID],[7,8] Suwimon Khusuwan,[9] Watthanapong Suphamongkholchaikul,[10] Parinya Chamnan,[10] Watcharapong Piyaphanee,[11] Huong Thi Lan Vu ![ORCID],[12] Yen Hai Nguyen,[12] Khanh Hong Nguyen,[12] Thach Ngoc Pham,[13] Quang Minh Le,[14] Vinh Hai Vu ![ORCID],[14] Duc Minh Chau,[15] Dung Em Thi Hoang Vo,[15] Elinor K Harriss,[16] Hindrik Rogier van Doorn,[3,12] Raph Leonardus Hamers ![ORCID],[2,3] Fabiana Lorencatto,[17] Lou Atkins,[17] Direk Limmathurotsakul ![ORCID][1,3,18]

**Correspondence to**
Dr Direk Limmathurotsakul;
direk@tropmedres.ac

## ABSTRACT

**Objective** Blood culture (BC) sampling is recommended for all suspected sepsis patients prior to antibiotic administration. We examine barriers and enablers to BC sampling in three Southeast Asian countries.

**Design** A Theoretical Domains Framework (TDF)-based survey, comprising a case scenario of a patient presenting with community-acquired sepsis and all 14 TDF domains of barriers/enablers to BC sampling.

**Setting** Hospitals in Indonesia, Thailand and Viet Nam, December 2021 to 30 April 2022.

**Participants** 1070 medical doctors and 238 final-year medical students were participated in this study. Half of the respondents were women (n=680, 52%) and most worked in governmental hospitals (n=980, 75.4%).

**Outcome measures** Barriers and enablers to BC sampling.

**Results** The proportion of respondents who answered that they would definitely take BC in the case scenario was highest at 89.8% (273/304) in Thailand, followed by 50.5% (252/499) in Viet Nam and 31.3% (157/501) in Indonesia (p<0.001). Barriers/enablers in nine TDF domains were considered key in influencing BC sampling, including 'priority of BC (TDF-goals)', 'perception about their role to order or initiate an order for BC (TDF-social professional role and identity)', 'perception that BC is helpful (TDF-beliefs about consequences)', 'intention to follow guidelines (TDF-intention)', 'awareness of guidelines (TDF-knowledge)', 'norms of BC sampling (TDF-social influence)', 'consequences that discourage BC sampling (TDF-reinforcement)', 'perceived cost-effectiveness of BC (TDF-environmental context and resources)' and 'regulation on cost reimbursement (TDF-behavioural regulation)'. There was substantial heterogeneity between the countries. In most domains, the lower (higher) proportion of Thai respondents experienced the barriers (enablers) compared with that of Indonesian and Vietnamese respondents. A range of suggested intervention types and policy options was identified.

## STRENGTHS AND LIMITATIONS OF THIS STUDY

⇒ The Theoretical Domains Framework-based survey comprehensively identified individual, sociocultural and environmental barriers and enablers to blood culture (BC) sampling across study countries.

⇒ A convenience sampling approach, distributing invitations in letters, emails, pamphlets and online social media platforms, through existing collaborations in hospitals in the three survey countries was used and might have led to selection bias.

⇒ The target sample size was not reached in Thailand.

⇒ The findings may not be generalisable to all low-income and middle-income countries because barriers and enablers to BC sampling can be varied and local evaluations are needed.

**Conclusions** Barriers and enablers to BC sampling are varied and heterogenous. Cost-related barriers are more common in more resource-limited countries, while many barriers are not directly related to cost. Context-specific multifaceted interventions at both hospital and policy levels are required to improve diagnostic stewardship practices.

## INTRODUCTION

Blood culture (BC) is a crucial diagnostic, which can guide antibiotic treatment decisions of severe bacterial infections, and may improve patient outcomes.[1 2] The cumulative results of BC are also crucial to inform antimicrobial resistance (AMR) surveillance, at the hospital, country and global levels.[3] International guidelines on sepsis management have been stressing the importance of obtaining

BC before or, when not possible, within 24 hours after administration of antibiotics.[1 4]

Nonetheless, BC is generally underused, both in high-income countries (HICs) and low-income and middle-income countries (LMICs), with wide variations in reported BC sampling rates between hospitals and global regions. Reported BC sampling rates ranged from 196 to 308 per 1000 patient-days in the USA,[5 6] from 6.7 to 86.5 per 1000 patient-days in the European Union,[7] from 0 to 82 per 1000 patient-days in the Central Asian and European Surveillance of AMR network[8] and 31, 82 and 10 per 1000 patient-days in selected hospitals in Indonesia,[9] Thailand[10] and Viet Nam,[11] respectively.

A range of barriers and enablers have been identified that influence BC sampling, based on different study designs, theories and frameworks. Lack of clear guidelines, training, microbiological infrastructure and positive attitudes regarding BC among medical practitioners are commonly reported barriers.[8 12–15]

Changing the behaviour of medical practitioners is complex, and a systematic approach has been shown useful to understand factors influencing adherence to guidelines or recommendations so as to inform the design of future interventions.[16–18] The Theoretical Domains Framework (TDF) has been developed by synthesising a wide range of theories and enables researchers to investigate a broader range of individual, sociocultural and environmental behavioural influences than they would with a single theory alone.[16–18] The TDF has been widely used to explore barriers and enablers to healthcare professional behaviours, including diagnostic testing, antimicrobial stewardship and infection prevention control.[19–22]

Here, we aimed to identify barriers and enablers to BC sampling in three middle-income countries in Southeast Asia (SEA) using a theory-based approach informed by the TDF.

## METHODS

### The TDF survey

We developed a TDF survey questionnaire, comprising a hypothetical case scenario and all 14 TDF domains of barriers/enablers to BC sampling, through an iterative process of systematic literature review and previous TDF surveys on other health topics (table 1; online supplemental appendix S1 and S2).[23–26] Each question used a five-point Likert scale representing the level of perceived barriers/enablers to BC sampling under all TDF domains.

The initial questionnaire was translated into Thai, Vietnamese and Indonesian language and piloted among 10–19 medical doctors and 3–6 final-year medical students in each country (a total of 54 respondents) to test the clarity of questions and choice answers in each language and to ensure no potential key barriers/enablers were omitted. We asked respondents to complete the survey and provide feedback using 1:1 interviews via phone or using online meeting software. The questionnaire was revised and finalised based on the pilot study results.

During the pilot survey, we included 'monetary reward' and 'monetary fine' as examples of positive and negative consequences to BC sampling, respectively. We received strong feedback that those are not present for BC sampling in Indonesia, Thailand and Viet Nam. Therefore, the word 'monetary reward' and 'monetary fine' were removed. One free-text question was added (ie, questions 6–5, 'additional comments about emotional factors…'), a total of 27 choice answers were added and languages and wordings were revised. The final questionnaire included 54 questions about barriers/enablers to BC sampling and respondents' demographic characteristics (online supplemental appendix S3).

### Study participants

We invited medical doctors and final-year medical doctors in Indonesia, Thailand and Viet Nam to complete the online TDF survey. We used a convenience sampling approach, distributing invitations in letters, emails, pamphlets and online social media platforms, through existing collaborations in hospitals in the three survey countries. The online cross-sectional survey was conducted using the Qualtrics survey platform. Multiple participation was prevented by using the Prevent Ballot Box Stuffing Option within Qualtrics.

We used a simple formula for calculating the sample size.[27] Assuming prevalence of a barrier or enabler to be 50% among medical doctors, with a margin of error 5%, the sample size of medical doctors was estimated to be at least 385 per country. Assuming prevalence of a barrier or enabler to be 50% among final-year medical students, with a margin of error 10%, the sample size of final-year medical students was estimated to be at least 97 per country. Therefore, we aimed to enrol 400 medical doctors and 100 final-year medical students in each country (a total of 1500 respondents).

### Analysis

For each question, we defined that respondents who answered 'definitely'/'likely', 'all the time'/'often' or 'strongly agree'/'agree' perceived the importance or agreement with that barrier/enabler. The proportion of respondents who answered likewise, after excluding respondents who answered 'I do not know' or 'I do not want to answer', was presented. Groups were compared by $\chi^2$ or Fisher exact tests as appropriate. Logistic regression models with random effects for countries, for hospital type nested in the same country and for professional roles nested in the same hospital type were used to evaluate the association between respondents' answers about each barrier/enabler and to the case scenario. Multivariable logistic regression model was not used because we considered that each key TDF domain could influence BC sampling practice via a causal relationship and should be addressed in future interventions. Statistical analyses were performed using Stata V.15.1 (StataCorp, USA).

We identified and ranked important TDF domains by scoring them based on an established set of four

**Table 1** Key questions for barriers and enablers to blood culture (BC) sampling

| TDF domains | Questions |
| --- | --- |
| Knowledge | Do you know of any recommendation(s) or guideline(s) for BC sampling being used in your hospital? |
| | Are you aware of any international recommendation(s) or guideline(s) for blood culture sampling? |
| | In your hospital, are there any training, lectures, classes or meetings that provide you knowledge about local/national/international guidelines for BC sampling? |
| Skills | In your current hospital setting, which types of professionals are tasked to draw blood from patients for BC? |
| | How skilled are you in drawing blood? |
| Social professional role and identity | In your current hospital setting, which types of professionals/staff can order BC? |
| | Do you think that it is an appropriate part of your current job to order BC? |
| | Do you think that it is an appropriate part of your current job to draw blood for BC? |
| Beliefs about capabilities | If you have to draw blood yourself, are you confident that you can draw blood successfully? 'Successfully' means obtaining blood. |
| | Are you confident that others (who are tasked to draw blood in your hospital) can draw blood successfully? |
| | Are you confident that you can draw blood appropriately? 'Appropriately' means that general recommendations for blood culture specimen collection such as aseptic technique are followed. |
| | Are you confident that others (who are tasked to draw blood in your hospital) can draw blood appropriately? |
| Optimism | In your current hospital setting, how optimistic are you that a BC will be sampled and processed in the laboratory appropriately if you order a BC? |
| Beliefs about consequences | Do you agree or disagree about the following potential advantages of BC, making BC helpful in your current hospital setting? |
| | Do you agree or disagree about the following disadvantages of BC, making BC unnecessary in your current hospital setting? |
| Reinforcement | Are there any positive consequences to you, if you order BC when recommended? |
| | Are there any negative consequences to you, if you do not order BC when recommended? |
| | Are there any negative consequences to you, if you order BC when recommended? |
| Intentions | How often do you plan to follow the recommendation(s) or guideline(s) for BC sampling being used in your hospital? |
| Goals | How often do you obtain BC prior to administration of empirical antibiotics in patients presenting with sepsis? |
| Memory, attention and decision processes | Apart from the recommendation(s) or guideline(s) being used at your hospital, do you have any additional reasons for deciding to do BC sampling? |
| | Would you still order blood culture in case patients are already on antibiotics? |
| | Would you still order blood culture in case patients have anaemia? |
| Environmental context and resources | Regardless of who pays for the cost of BC, would you say that the benefits of BC outweigh the cost? |
| | How often do patients have to pay for BC using their own money (ie, out of pocket)? |
| | Do you consider whether patients can afford the cost of BC as a reason for deciding to do BC sampling? |
| | In your hospital, how often could you not order BC because consumables (such as blood culture bottles, needles, syringes, blood collection set, etc) are not available? |
| Social influences | To what extent do you order BC sampling because you are following local norms? 'Norms' mean usual practice that are typical of or accepted within your hospital. |
| | Do following people (such as consultants, head of the department, executives of the hospital, patients and family of patients) have any positive or negative influence on you to order BC? |
| Emotion | Apart from your logical considerations, do you think that any emotional factors of anyone are involved in ordering and sampling for BC? |
| Behavioural regulation | In your hospital, are there any procedures that support you to order or regulate ordering of BC per local/national/international guidelines? |
| | Do you consider whether patients have a health scheme or insurance that covers the cost of BC as a reason for deciding to do BC sampling? |

TDF, Theoretical Domain Framework.

**Table 2** Links between TDF and COM-B components*

| COM-B components | | TDF domains |
|---|---|---|
| Capability | Psychological | Knowledge |
| | | Skills |
| | | Memory, attention and decision processes |
| | | Behavioural regulation |
| | Physical | Skills |
| Opportunity | Social | Social influences |
| | Physical | Environmental context and resources |
| Motivation | Reflective | Social/professional role and Identity |
| | | Beliefs about capabilities |
| | | Optimism |
| | | Beliefs about consequences |
| | | Intentions |
| | | Goals |
| | Automatic | Social/professional role and Identity |
| | | Optimism |
| | | Reinforcement |
| | | Emotion |

*COM-B component stands for Capability (physical capability or psychological capability), Opportunity (physical opportunity or social opportunity), Motivation (automatic motivation or reflective motivation)–Behaviour, represents source of the behaviours and is the core of the Behaviour Change Wheel.[16–18]
TDF, Theoretical Domains Framework.

'importance criteria' (modified from a previous TDF study[28]): (1) 'frequency' (the proportion of respondents who perceived the importance or agreement with a barrier/enabler); (2) 'elaboration' (number of themes within each domain); (3) 'expressed importance' (quotes from respondents expressing importance or agreement); and (4) 'association between reported barriers/enablers and BC practice' (size of effect and strength of association, ie, ORs and p values, obtained from the logistic regression models, respectively). P values<0.05 were not used as a simple cut-off whether an association was present or absent.[29 30] P values<0.001 were regarded as providing strong evidence against the null hypothesis. For a negative association (OR<1.0), the inversed OR (1/OR) was considered as the size effect when compared with other positive associations. Overall rank was decided based on detailed presentation of the ratings of each criterion.

Lastly, we mapped identified TDF domains to the 'Capability', 'Opportunity', 'Motivation' and 'Behaviour' (COM-B) model (table 2).[16–18] COM-B forms the hub of the Behaviour Change Wheel (BCW), a framework which signposts to potentially relevant intervention strategies. This allowed us to list all intervention types and policy options that were likely to be effective in addressing identified barriers and enablers.

**Patient and public involvement**
None.

## RESULTS

From 1 December 2021 to 30 April 2022, 1070 medical doctors and 238 final-year medical students in Indonesia, Thailand and Viet Nam completed the online TDF survey. Half of respondents were women (n=680, 52%) and most worked in governmental hospitals (n=980, 75.4%) (table 3 and online supplemental appendix S4). The most common department was internal medicine (n=450, 34.4%), followed by emergency (n=175, 13.4%) and paediatrics (n=153, 11.7%). Respondents were from 24 of 34 provinces in Indonesia, 39 of 77 provinces in Thailand and 25 of 63 provinces in Viet Nam.

Based on the case scenario of a patient presenting with community-acquired sepsis, half of respondents (52.3%, 682/1304) answered that they would definitely take BC. However, the responses were significantly different between the three countries (p<0.001). Most Thai respondents (89.8%, 273/304) answered that they would definitely take BC, while half of Vietnamese respondents (50.5%, 252/499) and about a third of Indonesian respondents (31.3%, 157/501) did.

Using an established set of four 'importance criteria', we ranked important TDF domains by scoring as shown in table 4. We present, in rank order, the nine TDF domains that were considered very important (ie, key) in the three countries in SEA in the section below.

### TDF-goals

TDF-goals domain covers mental representations of outcomes that an individual wants to achieve, goal priority and implementation intention.[16–18]

#### Theme: priority of BC

In many settings, ordering or initiating an order for BC can take only few seconds by writing 'blood culture' in the doctor order form. We used a question asking about the priority of BC compared with that of empirical antibiotics, and 91.3% (274/300) of Thai respondents answered that they obtain BC prior to administration of empirical antibiotics all the time or often, while 80.0% (380/475) of Vietnamese respondents and 54.2% (251/463) of Indonesian respondents answered likewise (p<0.001, online supplemental appendix S4). Respondents who gave priority to BC were more likely to answer with 'definitely take BC' in the case scenario (OR 4.25, 95% CI 3.04 to 5.94, p<0.001, online supplemental appendix S6). Example quotes related to the priority of BC were "If other urgent examinations are to be required, BC could be delayed (Vietnamese respondent [barrier])" and "BC should be performed, although the results are often negative.

**Table 3** Demographics and responses to the hypothetical case scenario

| Variables | Indonesia (n=503) | Thailand (n=304) | Viet Nam (n=501) | P values |
|---|---|---|---|---|
| Female gender | 263 (52.3%) | 195 (64.1%) | 222 (44.3%) | <0.001 |
| Hospital types | | | | |
| Government hospital | 340 (67.6%) | 209 (68.8%) | 431 (86.0%) | <0.001 |
| Private hospital | 113 (22.5%) | 15 (4.9%) | 17 (3.4%) | |
| University hospital | 26 (5.2%) | 76 (25.0%) | 29 (5.8%) | |
| Other* | 19 (3.8%) | 2 (0.7%) | 22 (4.4%) | |
| I do not want to answer | 5 (1.0%) | 2 (0.7%) | 2 (0.4%) | |
| Hospital bed size | | | | |
| <200 | 99 (19.7%) | 35 (11.5%) | 24 (4.8%) | <0.001 |
| 201–400 | 107 (21.3%) | 46 (15.1%) | 29 (5.8%) | |
| 401–600 | 72 (14.3%) | 39 (12.8%) | 62 (12.4%) | |
| 601–1000 | 66 (13.1%) | 45 (14.8%) | 144 (28.7%) | |
| 1001–2000 | 39 (7.8%) | 82 (27.0%) | 125 (25.0%) | |
| >2000 | 27 (5.4%) | 30 (9.9%) | 74 (14.8%) | |
| I do not know | 89 (17.7%) | 27 (8.9%) | 35 (7.0%) | |
| I do not want to answer | 4 (0.8%) | 0 (0%) | 8 (1.6%) | |
| Current job† | | | | |
| Medical doctor—executive level | 13 (2.6%) | 5 (1.6%) | 17 (3.4%) | <0.001 |
| Medical doctor—consultant level | 74 (14.7%) | 75 (24.7%) | 198 (39.5%) | |
| Medical doctor—physician level | 124 (24.7%) | 38 (12.5%) | 112 (22.4%) | |
| Medical doctor—resident level | 168 (33.4%) | 63 (20.7%) | 101 (20.2%) | |
| Medical doctor—intern level | 33 (6.6%) | 35 (11.5%) | 14 (2.8%) | |
| Final-year medical student | 91 (18.1%) | 88 (28.9%) | 59 (11.8%) | |
| Department | | | | |
| Internal medicine | 149 (29.6%) | 155 (51.0%) | 146 (29.1%) | <0.001 |
| Paediatrics | 65 (12.9%) | 43 (14.1%) | 45 (9.0%) | 0.05 |
| Infection disease division/department | 12 (2.4%) | 5 (1.6%) | 56 (11.2%) | <0.001 |
| Surgery | 21 (4.2%) | 45 (14.8%) | 81 (16.2%) | <0.001 |
| Orthopaedics | 6 (1.2%) | 18 (5.9%) | 14 (2.8%) | 0.001 |
| Obstetrics/gynaecology | 20 (4.0%) | 29 (9.5%) | 7 (1.4%) | <0.001 |
| Emergency department | 112 (22.3%) | 34 (11.2%) | 29 (5.8%) | <0.001 |
| Intensive care unit | 45 (8.9%) | 13 (4.3%) | 51 (10.2%) | 0.01 |
| Would you take a blood culture sample in the hypothetical case scenario (presenting with community-acquired sepsis)?‡ | | | | |
| Definitely (>95%–100% of the time) | 157 (31.2%) | 273 (89.8%) | 252 (50.3%) | <0.001 |
| Likely (75%–95% of the time) | 138 (27.4%) | 23 (7.6%) | 149 (29.7%) | |
| Maybe (25%–74% of the time) | 116 (23.1%) | 5 (1.6%) | 70 (14.0%) | |
| Unlikely (5%–24% of the time) | 44 (8.7%) | 2 (0.7%) | 19 (3.8%) | |
| Rarely (ranging from never to <5% of the time) | 46 (9.1%) | 1 (0.3%) | 9 (1.8%) | |
| I do not know | 1 (0.2%) | 0 (0%) | 1 (0.2%) | |
| I do not want to answer | 1 (0.2%) | 0 (0%) | 1 (0.2%) | |

*Included clinics (n=3) and text answers that could not be used to determine the hospital type such as internship and medical students.
†In the survey, for a medical doctor, 'executive level' was defined as having an administrative position without clinical work, 'consultant' was defined as having a clinical specialty degree, 'resident' as currently under postgraduate clinical training, 'physician' as having no clinical specialty/subspecialty degree and not under postgraduate clinical training and 'intern' as a recent medical school graduate in the first year of postgraduate on-the-job training.
‡Hypothetical case scenario. 'A 72-year-old woman who was brought to the emergency department of your hospital by her daughter when she noticed the patient was more confused than her baseline and was found to have a high fever and fast breathing. She had an auscultatory finding compatible with pneumonia. It is decided that this patient will be admitted to your hospital'. If you have an authority to take a blood culture, would you take blood culture sample(s) in this case on admission?

**Table 4** Criteria and rank of TDF domains regarding barriers/enablers to BC sampling

| TDF domains | (1) 'Frequency' or the proportion of respondents who perceived the importance or agreement with a barrier/enabler within each domain* | (2) 'Elaboration' or number of themes within each domain† | (3) 'Expressed importance' or quotes from respondents expressing importance or agreement with a barrier/enabler within each domain‡ | (4) 'Association between reported barriers or enablers and BC practice' or size of effect and strength of association, that is, OR and p values, obtained from the logistic regression model, respectively§ | Overall rank¶ |
|---|---|---|---|---|---|
| Goals | Moderate (25%–74%) | 1 | A few quotes | OR 4.25, strongly associated | Very important |
| Social professional role and identity | High (75%–95%) | 3 | A few quotes | OR 3.36, strongly associated | Very important |
| Beliefs about consequences | High (75%–95%) | 2 | A number of quotes | OR 2.96, strongly associated | Very important |
| Intentions | Moderate (25%–74%) | 1 | A few quotes | OR 2.92, strongly associated | Very important |
| Knowledge | Moderate (25%–74%) | 2 | A few quotes | OR 2.55, strongly associated | Very important |
| Social influences | Moderate (25%–74%) | 2 | A number of quotes | OR 2.20, strongly associated | Very important |
| Reinforcement | Moderate (25%–74%) | 2 | A number of quotes | OR 0.48, strongly associated | Very important |
| Behavioural regulation | Moderate (25%–74%) | 2 | A number of quotes | OR 1.65, strongly associated | Very important |
| Environmental context and resources | High (75%–95%) | 3 | A number of quotes | OR 1.63, strongly associated | Very important |
| Emotion | Low (5%–24%) | 2 | A number of quotes | Not observed | Important |
| Optimism | High (75%–95%) | 1 | None | OR 1.78, strongly associated | Important |
| Skills | Moderate (25%–74%) | 1 | None | OR 1.74, associated | Important |
| Memory, attention and decision processes | Moderate (25%–74%) | 2 | A few quotes | Not observed | Important |
| Beliefs about capabilities | Moderate (25%–74%) | 2 | None | Not observed | Important |

*For each question, we defined that respondents who answered 'definitely'/'likely', 'all the time'/'often' or 'strongly agree'/'agree' perceived the importance or agreement with that barrier/enabler. The highest proportion for a barrier/enabler in each domain is presented. Details are presented in the online supplemental appendix S4.
†Additional details are presented in the online supplemental appendix S1.
‡Details are presented in the online supplemental appendix S5.
§Details are present in the online supplemental appendix S6.
¶Overall rank was decided based on detailed presentation of the ratings of each criterion.
BC, blood culture; TDF, Theoretical Domains Framework.

We can't wait for patients not responding to empirical antibiotics before starting BC (Indonesian respondent [enabler])" (online supplemental appendix S5).

### TDF-social professional role and identity
#### Theme: perception about their role to order or initiate an order for BC
Most medical doctors (86.5%, 905/1046) answered that it is very appropriate or appropriate for them to order BC or initiate an order for BC, while only about half of final-year medical students (49.8%; 115/231) answered likewise (p<0.001). Among medical doctors, 95.8% (207/216) of Thai respondents answered that it is very appropriate or appropriate for them to order BC or initiate an order for BC, while 87.0% (368/423) of Vietnamese respondents and 81.1% (330/407) of Indonesia respondents answered likewise (p<0.001). The respondents who answered that it is their role to order or initiate an order for BC were

more likely to answer with 'definitely take BC' in the case scenario (OR 3.36, 95% CI 2.50 to 4.51, p<0.001).

#### Theme: level of doctors who can order or initiate an order for BC
More than 75% of Thai respondents answered that all levels of medical doctors (consultants, physicians, residents and interns) can order or initiate an order for BC in their hospitals, while most Indonesian and Vietnamese respondents (87.9%, 870/990) answered that consultants can, but fewer answered that physicians (61.8%, 612/990), residents (59.1%, 585/990) and interns (20.3%, 201/990) can (p<0.001). A quarter of Thai respondents (28.7%, 87/303) answered that final-year medical students can order or initiate an order for BC under supervision of attending medical doctors, while Indonesian respondents (2.2%, 11/500) and Vietnamese respondents (0.6%, 3/490) rarely answered

likewise (p<0.001). None reported that nurses can order or initiate an order for BC.

### Theme: perception about their role to draw blood for BC

Most respondents (72.8%, 949/1303) answered that registered nurses are tasked to draw blood from patients for BC, followed by microbiology laboratory team (36.0%, 469/1303), specialised blood draw team (27.4%, 357/1303), residents (25.4%, 331/1303), physicians (23.5%, 306/1303), consultants (23.2%, 302/1303), interns (17.8%, 229/1303) and final-year medical students (11.6%, 151/1303). Of respondents who answered that they are tasked to draw blood for BC themselves, 69.1% (248/359) responded that it is very appropriate or appropriate for their role to draw blood for BC. Those respondents were more likely to answer with 'definitely take BC' in the case scenario (OR 1.94, 95% CI 1.04 to 3.64, p=0.04).

### TDF-belief about consequences
#### Theme: perceived that BC is helpful

Most respondents strongly agreed or agreed that BC is helpful in adjusting antibiotics (94.0%, 1224/1302), clinical decisions (93.6%, 1220/1303), detecting AMR bacterial infections (92.1%, 1199/1302), ruling in an infection (90.2%, 1172/1299), reducing overuse of antibiotics (87.4%, 1140/1304) and reducing patient mortality (79.2%, 1027/1297). Most respondents strongly agreed or agreed that accumulative results of BC are helpful in understanding epidemiology of AMR bacterial infections (94.5%, 1228/1299). More than half of respondents strongly agreed or agreed that BC is helpful in reducing length of hospital stay (72.3%, 938/1298) and ruling out an infection (60.5%, 786/1300).

Respondents who perceived that BC is helpful in clinical decisions (OR 2.96, 95% CI 1.71 to 5.12, p<0.001), reducing patient mortality (OR 1.61; 95% CI 1.18 to 2.20, p=0.003), ruling in an infection (OR 1.58, 95% CI 1.04 to 2.39, p=0.03), reducing length of hospital stay (OR 1.53, 95% CI, 1.14 to 2.04, p=0.004) or understanding epidemiology of AMR bacterial infections (OR 2.89, 95% CI 1.60 to 5.19, p<0.001) were more likely to answer with 'definitely take BC' in the case scenario. The proportion of respondents who answered that BC is helpful in clinical decisions was highest in Thai (97.7%, 297/304), followed by Indonesia (96.6%, 483/500) and Viet Nam (88.2%, 440/499, p<0.001).

#### Theme: perceived that BC is unnecessary

Some respondents strongly agreed or agreed that BC is unnecessary because it is not too late to collect BC later, particularly if patients do not improve after receiving empirical antibiotic treatment (32.7%, 423/1293), the therapeutic consequence of BC sampling is questionable (18.6%, 238/1277), antibiotic therapy can be determined based on clinical presentations (17.5%, 228/1301), results are often delayed (17.0%, 220/1298) quality of laboratory is questionable (15.3%, 194/1269), the scientific basis of the guideline on BC is questionable (15.0%, 191/1277), results are often negative or no growth (11.4%, 148/1295) and results are often contaminated (11.1%, 143/1288).

Respondents who perceived that BC is unnecessary because BC is not benefiting the patients (OR 0.37; 95% CI 0.24 to 0.57, p<0.001), it is not too late to collect BC later, particularly if patients do not improve after receiving empirical antibiotic treatment (OR 0.37; 95% CI 0.27 to 0.52, p<0.001), BC results are often delayed (OR 0.48, 95% CI 0.33 to 0.69, p<0.001), quality of laboratory is questionable (OR 0.48; 95% CI 0.33 to 0.70, p<0.001), antibiotic therapy can be determined based on clinical presentation (OR 0.51, 95% CI 0.36 to 0.73, p<0.001), a contaminated result often leads to wrong therapeutic approach (OR 0.53; 95% CI 0.30 to 0.95, p=0.03), BC results are often not interpretable (OR 0.54, 95% CI 0.34 to 0.87, p=0.01), BC results are often negative or no growth (OR 0.58, 95% CI 0.39 to 0.88, p=0.01), levels of local antibiotic resistance are low (OR 0.64; 95% CI 0.41 to 0.98, p=0.04), cultures are often contaminated (OR 0.64, 95% CI 0.42 to 0.98, p=0.04) and the scientific basis of the guideline on BC is questionable (OR 0.66, 95% CI 0.45 to 0.98, p=0.04) were less likely to answer with 'definitely take BC' in the case scenario. The proportion of respondents who answered that BC is not benefitting the patients was not different between countries (5.9%, 76/1297, p=0.38).

### TDF-intention

TDF-intention domain covers a conscious decision to perform or a resolve to act in a certain way, and stability of intentions.[16–18]

#### Theme: intention to follow guidelines

Among those who answered that they know of local guidelines, 92.9% (157/169) of Thai respondents answered that they plan to follow local guidelines all the time or often, while 82.0% (283/345) of Vietnamese respondents and 74.1% (172/232) of Indonesian respondents answered likewise (p<0.001). Respondents who intended to follow local guidelines were more likely to answer with 'definitely take BC' in the case scenario (OR 2.92, 95% CI 1.88 to 4.53, p<0.001).

### TDF-knowledge
#### Theme: awareness of guidelines

The proportion of respondents who answered that they know of local guidelines for BC sampling was highest in Viet Nam (70.7%; 347/491), followed by Thailand (56.3%, 169/300) and Indonesia (48.9%, 240/503, p<0.001). The proportion of respondents who answered that they know of international guidelines for BC sampling (47.8%, 596/1248) was not different between countries (p=0.73). Respondents who answered that they know of local guidelines (OR 2.55, 95% CI 1.93 to 3.38, p<0.001) or international guidelines (OR 1.97, 95% CI 1.50 to 2.57, p<0.001) were more likely to answer with 'definitely take BC' in the case scenario.

## Theme: training

The proportion of respondents who answered that there were no training, lectures, classes or meetings that provide knowledge about local/national/international guidelines for BC sampling in their hospitals was highest in Indonesia (37.8%, 153/407), followed by Thailand (24.9%, 64/257) and Viet Nam (12.5%, 52/421, p<0.001). Respondents who answered that there are training, lectures, classes or meetings that provide knowledge about guidelines for BC sampling were more likely to answer with 'definitely take BC' in the case scenario (OR 1.68; 95% CI 1.18 to 2.38, p=0.004).

## TDF-social influence
### Theme: norms of BC sampling

Most Thai respondents (78.5%, 233/297) answered that they order BC because they are following local norms all the time or often, while 51.5% (238/462) of Vietnamese respondents and 43.8% (180/411) of Indonesian respondents answered likewise (p<0.001). The respondents who answered that they order BC because they are following local norms were more likely to answer with 'definitely take BC' in the case scenario (OR 2.20, 95% CI 1.67 to 2.90, p<0.001).

### Theme: influences from healthcare workers, patients and family of patients

Most respondents (79.4%) answered that there are very positive or positive influences on BC sampling from consultants, followed by residents (64.5%), doctors (64.6%), heads of department (65.9%), executive levels (50.6%), nurses (47.6%), interns (45.2%), patients (43.0%) and family of patients (31.9%). Some respondents said that there are negative or very negative influence in BC sampling from family of patients (6.8%), nurses (5.2%), patients (4.3%) and executives of the hospital (3.6%). A number of quotes on this theme were noted; including "Negative influence in the order of BC is cost. Supervisor or the executives (of the hospitals) gave an order to control the cost (Thai respondent [barrier])" and "Sometimes, when the blood puncture fails on the first try, patients and their families refuse to have more blood drawn (Indonesian respondent [barrier])" (online supplemental appendix S5).

## TDF-reinforcement
### Theme: consequences that discourage BC sampling

Some respondents (32.5%, 300/923) answered that, if they order a BC when it is recommended, there are either negative social consequences (eg, verbal reprimand or any pressure from supervisors/executives of the hospital as the hospital (may) have to pay for the (extra) cost of BC) or negative material consequences (eg, a negative score, that doctors are at risk of having to spend extra time and effort to reimburse the cost of BC from any health scheme or insurance, or that doctors are at risk of having to pay for the (extra) cost of BC themselves). The proportion of those who answered likewise was highest

in Viet Nam (42.2%, 153/363), followed by Thailand (27.0%, 60/222) and Indonesia (25.7%, 87/338). Those who answered that there are negative consequences were less likely to answer with 'definitely take BC' in the case scenario (OR 0.48; 95% CI 0.34 to 0.67, p<0.001). A number of quotes on this theme were noted; including 'Warnings are given due to the costly examination, especially for patients insured with the Healthcare and Social Security Agency (Indonesian respondent [barrier])' and "Sometimes, the cost of BC cannot be reimbursed, and the doctor has to pay (Vietnamese respondent [barrier])" (online supplemental appendix S5).

## TDF-behavioural regulation
### Theme: regulation of cost reimbursement

Some respondents stated that 'whether patients have a health scheme or insurance that covers the cost of BC' (15.0%, 196/1308) and that 'whether patients are likely to have a final diagnosis that includes the cost of BC in the package of fee for service' are their additional reasons for deciding to order BC (11.6%, 152/1308). Those respondents were not associated with answering with 'definitely take BC' in the case scenario (p>0.20, both). However, a number of quotes on this theme were noted; including "The insurance often disapproves of BC examination. It is only approved when patients are admitted to the ICU or HCU [High Care Unit] (Indonesian respondent [barrier])" and "Medical professionals often object to BC due to tiredness [disheartened feeling] and the consequence of reduced reimbursement (Vietnamese respondent [barrier])" (online supplemental appendix S5).

### Theme: procedures to support or regulate doctors to order BC

Overall, the most common procedures to support or regulate doctors to order BC in respondents' hospitals were case reviews (eg, grand rounds or morning ward rounds, and BC is often mentioned; 30.8%, 326/1060), followed by standard order forms to remind ordering BC (29.9%, 317/1060), stewardship programmes and reviewing BC is included in the programmes (19.5%, 207/1060), posters (15.4%, 163/1060) and computer systems to remind ordering BC (10.7%, 113/1060). Respondents who answered that there were case reviews (OR 1.55, 95% CI 1.14 to 2.13, p=0.006) or stewardship programmes (OR 1.65, 95% CI 1.16 to 2.34, p=0.005) were more likely to answer with 'definitely take BC' in the case scenario

## TDF-environmental context and resources
### Theme: perceived cost-effectiveness of BC

Most Vietnamese respondents (85.9%, 407/474) considered that BC is very likely or likely to be cost-effective, while 79.5% (232/292) of Thai respondents and 68.8% (311/452) of Indonesian respondents considered likewise. The respondents who considered that BC is cost-effective were more likely to answer with 'definitely take BC' in the case scenario (OR 1.63, 95% CI 1.17 to 2.26, p<0.001).

**Theme: availability of microbiology laboratories, transport modalities, resources and consumables**

Some respondents answered that they could not order BC because microbiology laboratories are not available or not functioning (13.4%, 157/1174) or consumables (such as BC bottles, needles, syringes, blood collection set, etc) are not available (12.7%, 150/1181) all the time or often. Those respondents were not associated with answering with 'definitely take BC' in the case scenario (p>0.20 both)

**Theme: out of pocket**

About a quarter of Indonesian respondents (23.3%, 78/335) answered that patients have to pay for BC using their own money (ie, out of pocket) all the time or often, while 12.2% (28/230) of Thai participant and 8.3% (34/408) of Vietnamese participant answered likewise (p<0.001). Those respondents were not associated with answering with 'definitely take BC' in the case scenario (p=0.29).

Additional results and the content themes in the domains that were not identified as key domains are described in online supplemental appendix S1. We observed that presence of many barriers/enablers was different between countries. However, the presence of those barriers/enablers was not strongly associated with the answer in the case scenario. For example, patients who are already on antibiotics. A quarter of Thai respondents (26.6%, 81/304) answered that they were very likely to still order BC, while only 14.4% (72/501) of Vietnamese respondents and 3.2% (16/503) of Indonesian respondents did (p<0.001). Those respondents were not associated with answering with 'definitely take BC' in the case scenario (p=0.13).

**Intervention types and policy options to improve BC sampling practice**

We used the links among TDF, COM-B and BCW and listed all suggested intervention types and policy options related to very important TDF domains in Indonesia,

**Table 5** Suggested intervention types and policy options to improve BC sampling practice based on very important TDF domains in Indonesia, Thailand and Viet Nam

| | COM-B components | | | | |
| --- | --- | --- | --- | --- | --- |
| | Psychological capability (TDF: knowledge and behavioural regulation) | Reflective motivation (TDF: goals, beliefs about consequence and intention) | Automatic motivation (TDF: reinforcement) | Physical opportunity (TDF: environmental context and resources) | Social opportunity (TDF: social influence) |
| **Intervention types*** | | | | | |
| Education | √ | √ | | | |
| Persuasion | | √ | √ | | |
| Incentivisation | | √ | √ | | |
| Coercion | | √ | √ | | |
| Training | √ | | | | |
| Restriction | | | | √ | √ |
| Environmental restructuring | | | √ | √ | √ |
| Modelling | | | √ | | |
| Enablement | √ | | √ | √ | √ |
| **Policy options*** | | | | | |
| Communication/marketing | | √ | √ | | |
| Guidelines | √ | √ | √ | √ | √ |
| Fiscal | √ | √ | √ | √ | √ |
| Regulation | √ | √ | √ | √ | √ |
| Legislation | √ | √ | √ | √ | √ |
| Environmental/social planning | √ | | √ | √ | √ |
| Service provision | √ | √ | √ | √ | √ |

*Suggested intervention types and policy options were identified using the links between TDF, the components of the COM-B and the Behaviour Change Wheel.[16–18]

BC, blood culture; COM-B, Capability, Opportunity, Motivation and Behaviour; TDF, Theoretical Domains Framework.

Thailand and Viet Nam (table 5 and online supplemental appendix S7). A range of potential strategies were identified. Some strategies target individual reinforcement, environmental structure and social influence (eg, providing an example for physicians to aspire to or imitate the BC sampling practice (intervention type-modelling) and increasing means and reducing barriers to increase capability and opportunity for all levels of doctors to order or initiate an order for BC (intervention type-enablement)). Some strategies operate at the policy or service provision level (eg, changing regulation of cost reimbursement (policy option-fiscal), development or implementation of local guidelines (policy option-guideline) and establishing rules or principles of BC practice (policy option-regulation)).

## DISCUSSION

Our study shows that barriers and enablers to BC sampling in SEA are varied and heterogenous. We consider that 'priority of BC (TDF-goals)', 'perception about their role to order or initiate an order for BC (TDF-social professional role and identity)', 'intention to follow guidelines (TDF-intention)', 'norms of BC sampling (TDF-social influence)', 'consequences that discourage BC sampling (TDF-reinforcement)' and 'regulation on cost reimbursement (TDF-behavioural regulation)' are key barriers/enablers. In Thailand,[10] where BC utilisation rate is relatively high compared with Indonesia[9] and Viet Nam,[11] the proportions of each enabler being reported by respondents is higher for many domains. For example, the proportion of respondents who gave priority to BC was highest in Thailand at 91.3%. Likewise, the proportions of each barrier being reported by Thai respondents is lower for many domains. For example, the proportion of respondents who answered that there are consequences that discourage BC sampling was highest in Viet Nam (42.2%) and the proportion of respondents who answered that patients have to pay for BC using their own money (ie, out of pocket) was highest in Indonesia (23.3%). To improve diagnostic stewardship practices, all stakeholders will need to consider all suggested intervention types and policy options and develop intervention content based on local context.[16–18]

'Priority to BC (TDF-goals)', 'perception about their role to order or initiate an order for BC (TDF-social professional role and identity)', 'intention to follow guidelines (TDF-intention)' and 'norms of BC sampling (TDF-social influence)' are likely key barriers to BC sampling in both HICs and other LMICs where resources for BC sampling are available to some extent.[8 12–15]

To our knowledge, 'priority of BC (TDF-goals)', 'level of doctors who can order or initiate an order for BC (TDF-social professional role and identity)' and 'influence from healthcare workers, patients and families of patients (TDF-social influence)' have never been evaluated in LMICs.[8 12–15] Those are important barriers/enablers. 'Priority of BC' has the highest OR for the association with

'definitely take BC' in the case scenario in our study (OR 4.25). The importance of 'priority of BC' was previously reported from HICs.[13] In addition, in many hospitals in both HICs and LMICs, final-year medical students and interns are responsible for most BC ordering and acquisition[31] and influences from other parties can discourage BC sampling.

Remarkably, the cost of BC seems to have influence on executive level doctors, patients, families of patients, medical doctors and those who set regulations on cost reimbursement of BC. This is shown by many quotes related to the cost of BC in the theme 'influences from healthcare workers, patients and family of patients (TDF-social influence)', 'consequences that discourage BC sampling (TDF-reinforcement)', 'perceived cost-effectiveness of BC (TDF-environmental context and resources)' and 'regulation on cost reimbursement (TDF-behavioural regulation)' (online supplemental appendix S5).

It is worth noting that the quotes related to the cost-related barriers are more common in Indonesian and Vietnamese respondents than in Thai respondents. Nonetheless, 'no priority of BC', 'lack of role to order BC', 'perceived that BC is unnecessary', 'no local guidelines for BC' and 'no intention to follow local guidelines' are examples of many barriers that are not directly related to cost.

To overcome cost-related barriers, multifacet interventions based on local context should be considered and implemented. For example, the interventions may include providing clear posters emphasising local guidelines for BC sampling over wide areas in hospitals (intervention type-environmental restructuring). This intervention type is aimed to increase social opportunity, physical opportunity and automatic motivation for medical doctors to adopt and practice the local guidelines for BC sampling (online supplemental appendix S7).[16–18] This intervention could reduce the barrier '(negative) influences from healthcare workers, patients and family of patients (TDF-social influence)' and 'perceived cost-effectiveness of BC (TDF-environmental context and resources)' if the importance and benefit of BC sampling are clearly present on the posters endorsed by the local hospitals and national authorities. Repeatedly announcing to all levels of healthcare workers that negative consequences that discourage BC sampling per local guidelines will not be tolerated (intervention type-enablement) could be considered and implemented to reduce the barrier '(negative) consequences that discourage BC sampling (TDF-reinforcement)'. Changing regulation of cost reimbursement and finding financial support for BC sampling per local guidelines (policy option-fiscal) could be considered and implemented to reduce the barrier 'regulation on cost reimbursement (TDF-behavioural regulation)'. Most importantly, multifacet interventions are recommended to be systematically designed based on barriers and enablers locally identified and based on local context.[16–18]

Fear of 'blood stealing' or 'blood selling' is reported as a barrier to blood specimen collection in many countries in sub-Saharan Africa; including Kenya, Zambia, Mozambique, The Gambia, Tanzania and Uganda.[32] We observed fears of pain, needles, drawing a lot of blood, anaemia, blood-transmitted diseases, etc (online supplemental appendix S5), but did not observe fear of 'blood stealing' or 'blood selling'. Emotional barriers to BC sampling are likely different depending on local regions.

This study has several limitations. First, we used a convenience sample of hospitals and practitioners, which might have led to selection bias. The sampling frame size and the response rate are unknown. It is possible that those who did not receive the invitation and those received the invitation but did not respond to the survey had different frequencies of or different barriers/enablers to BC sampling than those who participated in the study. This limited our ability to draw definite conclusions on the contemporary situation on barriers/enablers to BC sampling in each country and in SEA. Second, the survey could not reach the target sample size in Thailand despite substantial efforts. The study might not have enough power to evaluate all barriers and enablers adequately. Third, the findings may not be generalisable to all LMICs because barriers and enablers to BC sampling can be varied and local evaluations are needed.

In conclusion, this comprehensive analysis using TDF gives information across the entire spectrum of behavioural influences of BC sampling. These results can help local healthcare providers and policy-makers to develop and implement interventions aiming to improve diagnostic stewardship practices.

**Author affiliations**
[1] Mahidol Oxford Tropical Medicine Research Unit, Faculty of Tropical Medicine, Mahidol University, Bangkok, Thailand
[2] Oxford University Clinical Research Unit Indonesia, Faculty of Medicine, Universitas Indonesia, Jakarta, Indonesia
[3] Centre for Tropical Medicine and Global Health, Nuffield Department of Medicine, Oxford University, Oxford, UK
[4] Prof. Dr. R. D. Kandou Central Hospital, Manado, Indonesia
[5] Dr. Iskak District Hospital, Tulungagung, Indonesia
[6] Pasar Minggu District Hospital, Jakarta, Indonesia
[7] Faculty of Medicine, Universitas Indonesia, Jakarta, Indonesia
[8] Department of Internal Medicine, Cipto Mangunkusumo National General Hospital, Jakarta, Indonesia
[9] Chiangrai Prachanukroh Hospital, Chiang Rai, Thailand
[10] Sunpasitthiprasong Hospital, Ubon Ratchathani, Thailand
[11] Hospital for Tropical Diseases, Faculty of Tropical Medicine, Mahidol University, Bangkok, Thailand
[12] Oxford University Clinical Research Unit, Ha Noi, Viet Nam
[13] National Hospital of Tropical Diseases, Hanoi, Viet Nam
[14] Viet Tiep Hospital, Hai Phong, Viet Nam
[15] Dong Thap Hospital, My Tan, Viet Nam
[16] Bodleian Health Care Libraries, University of Oxford, Oxford, UK
[17] Centre for Behaviour Change, University College London, London, UK
[18] Department of Tropical Hygiene, Faculty of Tropical Medicine, Mahidol University, Bangkok, Thailand

**Acknowledgements** The authors thank the respondents and staff at all the study hospitals and Mahidol-Oxford Tropical Medicine Research Unit. The authors thank Surveillance and Epidemiology of Drug resistant Infections Consortium (SEDRIC) for the support and comments on the study. We thank Le Nguyen Minh Hoa, Vu Minh Duy, Dam Thi Hong Hanh and Hoang Bao Long for support in data collection.

**Contributors** FL, LA and DL designed and supervised the study. PS, KSA, RL, HTLV, HRvD and RLH participated in project design and facilitated data collection. AT, LWAR, RB, EJN, DUN, SK, WS, PC, WP, YHN, TNP, QML, VHV, DMC, DETHV and EKH facilitated data collection. PS analysed the data and wrote the first draft of the manuscript. All authors contributed to the writing or revision of the manuscript. PS and DL verified the data. DL is the guarantor.

**Funding** This research was funded by the Wellcome Trust (220557/Z/20/Z).

**Disclaimer** The views expressed are those of the author(s) and not necessarily those of the funders.

**Competing interests** None declared.

**Patient and public involvement** Patients and/or the public were not involved in the design, or conduct, or reporting, or dissemination plans of this research.

**Patient consent for publication** Not applicable.

**Ethics approval** This study involves human participants. The study was approved by the Oxford University Tropical Research Ethics Committee (OXTREC545-21) and local ethical committees at Iskak Tulungagung Hospital (070/7303/407.206/2021), Prof. Dr. R. D. Kandou Hospital (156/EC/KEPK-KANDOU/IX/2021), Pasar Minggu Hospital (EOCRU/RCH.216/10.2021/1145) in Indonesia, The National Hospital for Tropical Diseases (14HDDD/NDTU) in Viet Nam, Faculty of Tropical Medicine, Mahidol University (TMEC21-069), Sunpasitthiprasong Hospital (065/64S) and Chiangrai Prachanukroh Hospital (CR 0032.102/EC023) in Thailand. Participants gave informed consent to participate in the study before taking part.

**Provenance and peer review** Not commissioned; externally peer reviewed.

**Data availability statement** Data are available upon reasonable request. All authors recognise the value of sharing individual level data. We aim to ensure that data generated from all our research are collected, curated, managed and shared in a way that maximises their benefit. Data underlying this publication are available upon request to the Mahidol Oxford Tropical Medicine Research Uni Data Access Committee at https://www.tropmedres.ac/units/moru-bangkok/bioethics-engagement/data-sharing.

**ORCID iDs**
Ralalicia Limato http://orcid.org/0000-0002-5306-3254
Erni J Nelwan http://orcid.org/0000-0003-4064-5412
Huong Thi Lan Vu http://orcid.org/0000-0002-9579-5576
Vinh Hai Vu http://orcid.org/0000-0001-6130-7864
Raph Leonardus Hamers http://orcid.org/0000-0002-5007-7896
Direk Limmathurotsakul http://orcid.org/0000-0001-7240-5320

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
