## [Reviewer comments · BMJ Open]

ARTICLE DETAILS

TITLE (PROVISIONAL)	Barriers and enablers to blood culture sampling in Indonesia, Thailand and Vietnam: a Theoretical Domains Framework-based survey
AUTHORS	Suntornsut, Pornpan; Asadinia, Koe Stella; Limato, Ralalicia; Tamara, Alice; Rotty, Linda W.A.; Bramanti, Rendra; Nusantara, Dwi U.; Nelwan, Erni J.; Khusuwan, Suwimon; Suphamongkholchaikul, Watthanapong; Chamnan, Parinya; Piyaphanee, Watcharapong; Vu, Huong; Hai Yen, Nguyen; Nguyen Hong, Khanh; Pham, Ngoc Thach; Le, Minh Quang; Vu, Hai Vinh; Duc, Chau Minh; Vo, Thi Hoang Dung Em; Harriss, Elinor; van Doorn, H. Rogier; Hamers, Raph Leonardus; Lorencatto, Fabiana; Atkins, Lou; Limmathurotsakul, Direk

VERSION 1 – REVIEW

REVIEWER	Haukka, Kaisa University of Helsinki, Dept Microbiology
REVIEW RETURNED	10-Oct-2023

GENERAL COMMENTS	Barriers and enablers to blood culture sampling in Indonesia, Thailand and Vietnam The authors used Theoretical Domains Frameworks to explore barriers and enablers to ordering blood culture (BC) from potential sepsis patients visiting a hospital in three Asian countries. The manuscript is well written in general, but some points are unclear to me and need clarification: 1. A key factor influencing BC sampling is “priority of BC [TDF-goals]”. It is mentioned in the abstract, results, discussion and yet it’s difficult to grasp what it means, since the naming does not support intuitive understanding of the term. Table 1 and Results section explain it, but the comments in Table S6 are very valuable to understand what it means in practice.2. Also otherwise, Table S6 is very informative, but maybe too long to be included in the main text. Eg. a quote from a Vietnamese respondent [barrier] makes much sense: “In patients who have already received antibiotics, BC is not meaningful.” This aspect could be discussed more, did others bring it up etc. Is it common in those countries that patients self-medicate themselves with antibiotics before coming to see a doctor? Could this be one explanation to why there is so big difference between the countries, shown in Table 2 “Case-study: Would you take a BC sample in a case presenting with community-acquired sepsis?”3. Table 1 is not very helpful, eg. it presents a question “How skilled
---

	are you in drawing blood?” As it becomes evident when reading the questionnaire results, it is mainly other healthcare workers than physicians who draw the blood samples. Consider expanding the Table 1 to contain more accurate information. 4. In discussing most of the TDF domains, percentages are used to explain the results. Why in TDF-Belief about consequences themes report Odds Ratios instead? Their meaning is difficult to interpret in this analysis. 5. Information in the lines 147-149 is very undetailed. Consider bringing the first part of the Table S8 to the main text to make it clear what are the “links”. 6. Lines 378-379: clarify what you mean. 7. Lines 404-407: clarify what you mean. 8. What is MORU Data Sharing Policy (appendix S3)? 9. Transfer Appendix S4 (Criteria and rank of TDF domains) to the main text before the detailed explanations of different TDF domains. It draws the results together very nicely. 10. Conclusion (Lines 33-36) could be more informative, maybe mentioning the major factors to each country. Eg. the financial factors were possibly more important in poorer countries. This topic should also be discussed at lines 396- 411 for each country separately, if there was a difference. If not, it is also good to mention.
--	--

VERSION 1 – AUTHOR RESPONSE

Reviewer: 1

Dr. Kaisa Haukka, University of Helsinki

Comments to the Author:

Barriers and enablers to blood culture sampling in Indonesia, Thailand and Vietnam

The authors used Theoretical Domains Frameworks to explore barriers and enablers to ordering blood culture (BC) from potential sepsis patients visiting a hospital in three Asian countries. The manuscript is well written in general, but some points are unclear to me and need clarification:

1. A key factor influencing BC sampling is “priority of BC [TDF-goals]”. It is mentioned in the abstract, results, discussion and yet it’s difficult to grasp what it means, since the naming does not support intuitive understanding of the term. Table 1 and Results section explain it, but the comments in Table S6 are very valuable to understand what it means in practice.

Answer: We are thankful for the comments. Due to the difficulty to understand the TDF-goals, the following sentences were added to the result section TDF-goal for clarity, “In the TDF, ‘goals’ domain covers mental representations of outcomes that an individual wants to achieve, goal priority and implementation intention.” and “Example quotes related to the priority of BC were “If other urgent examinations are to be required, BC could be delayed (barrier).” and “BC should be performed, although the results are often negative. We can’t wait for patients not responding to empirical antibiotics before starting BC (enabler).” (Appendix S5).”

In addition, the following sentence was also added to the result section TDF-intention for clarity, “TDF-intention domain covers a conscious decision to perform or a resolve to act in a certain way, and stability of intentions.16-18”

2. Also otherwise, Table S6 is very informative, but maybe too long to be included in the main text. Eg. a quote from a Vietnamese respondent [barrier] makes much sense: “In patients who have already received antibiotics, BC is not meaningful.” This aspect could be discussed more, did others bring it up etc. Is it common in those countries that patients self-medicate themselves with antibiotics before coming to see a doctor? Could this be one explanation to why there is so big difference between the countries, shown in Table 2 “Case-study: Would you take a BC sample in a case presenting with community-acquired sepsis?”

Answer: We are thankful for the comments.

Based on the systematic literature review, we designed the survey and specifically evaluated this point carefully. Nonetheless, we found that “Patients who are already on antibiotics” were not associated with answering with “definitely take BC” in the case scenario ($p=0.13$). This was included in the ‘TDF-memory, attention and decision process’ (Supplementary Text).

In Indonesia, Thailand and Vietnam, it is not uncommon that patients who present with community-acquired sepsis at the hospitals had taken oral antibiotics or parenteral antibiotics prior to the hospital admission. To our knowledge, there is no previous strong evidence suggesting that proportion of previous exposure to antibiotics prior to hospital admission among patients presenting with sepsis is different between the three countries. Nonetheless, our study suggests that the proportion of Thai respondents who still order blood culture in this situation is higher than that of Vietnam and Indonesia respondents.

For clarity on this important point, the following sentences have been added to the result section and discussion as follows, “Additional results and the content themes in the domains that were not identified as key domains are described in Appendix S1. We observed that presence of many barriers/enablers was different between countries. However, the presence of those barriers/enablers was not strongly associated with the answer in the case scenario. For example, patients who are already on antibiotics. A quarter of Thai respondents (26.6%, 81/304) answered that they were very likely to still order BC, while only 14.4% (72/501) of Vietnamese respondents and 3.2% (16/503) of Indonesian respondents did ($p<0.001$). Those respondents were not associated with answering with “definitely take BC” in the case scenario ($p=0.13$).”

3. Table 1 is not very helpful, eg. it presents a question “How skilled are you in drawing blood?” As it becomes evident when reading the questionnaire results, it is mainly other healthcare workers than physicians who draw the blood samples. Consider expanding the Table 1 to contain more accurate information.

Answer: Table 1 has been expanded as suggested. Selected key quotes from appendix S5 were also included in result section.

4. In discussing most of the TDF domains, percentages are used to explain the results. Why in TDF-Belief about consequences themes report Odds Ratios instead? Their meaning is difficult to interpret in this analysis.

Answer: We presented only OR in the TDF-Belief about consequences to avoid repetitive presentations of all percentages from each question. We expanded the paragraph for TDF-Belief about consequence for clarity.

5. Information in the lines 147-149 is very undetailed. Consider bringing the first part of the Table S8 to the main text to make it clear what are the “links”.

Answer: The first part of the previous Table S8 has been moved to the main text as suggested.

6. Lines 378-379: clarify what you mean.

Answer: The sentence has been revised and expanded for clarity as follows, “In Thailand, where BC utilization rate is relatively high compared to Indonesia and Vietnam, the proportions of each enabler being reported by respondents is higher for many domains. For example, the proportion of respondents who gave priority to BC was highest in Thailand at 91.3%. Likewise, the proportions of each barrier being reported by Thai respondents is lower for many domains. For example, the proportion of respondents who answered that there are consequences that discourage BC sampling was highest in Vietnam (42.2%) and the proportion of respondents who answered that patients have to pay for BC using their own money (i.e. out of pocket) was highest in Indonesia (23.3%).”

7. Lines 404-407: clarify what you mean.

Answer: The paragraph has been revised and expanded for clarity as follows, “To overcome cost-related barriers, multi-facet interventions based on local context should be considered and implemented. For example, the interventions may include providing clear posters emphasizing local guidelines for BC sampling over wide areas in hospitals [Intervention type-environmental restructuring]. This intervention type is aimed to increase social opportunity, physical opportunity and automatic motivation for medical doctors to adopt and practice the local guidelines for BC sampling (Appendix S7). This intervention could reduce the barrier ‘(negative) influences from healthcare workers, patients and family of patients [TDF-social influence]’ and ‘perceived cost-effectiveness of BC [TDF-environmental context and resources]’ if the importance and benefit of BC sampling are clearly present on the posters endorsed by the local hospitals and national authorities. Repeatedly announcing to all levels of healthcare workers that negative consequences that discourage BC sampling per local guidelines will not be tolerated [Intervention type-enablement] could be considered and implemented to reduce the barrier ‘(negative) consequences that discourage BC sampling [TDF-reinforcement]’. Changing regulation of cost reimbursement and finding financial support for BC sampling per local guidelines [Policy option-fiscal] could be considered and implemented to reduce the barrier ‘regulation on cost reimbursement [TDF-behavioural regulation]’. Most importantly, multi-facet interventions are recommended to be systematically designed based on barriers and enablers locally identified and based on local context.”

8. What is MORU Data Sharing Policy (appendix S3)?

Answer: The ‘data sharing’ statement has been revised as follows, “Data are available upon reasonable request. All authors recognise the value of sharing individual level data. We aim to ensure that data generated from all our research are collected, curated, managed and shared in a way that maximises their benefit. Data underlying this publication are available upon request to the Mahidol Oxford Tropical Medicine Research Uni Data Access Committee at <https://www.tropmedres.ac/units/moru-bangkok/bioethics-engagement/data-sharing>.”

9. Transfer Appendix S4 (Criteria and rank of TDF domains) to the main text before the detailed explanations of different TDF domains. It draws the results together very nicely.

Answer: The previous Appendix S4 was moved to Table 3 in the main text as suggested.

10. Conclusion (Lines 33-36) could be more informative, maybe mentioning the major factors to each country. Eg. the financial factors were possibly more important in poorer countries. This topic should also be discussed at lines 396- 411 for each country separately, if there was a difference. If not, it is also good to mention.

Answer. The result and conclusion in the abstract has been revised and expanded as follows, “In most domains, the lower (higher) proportion of Thai respondents experienced the barriers (enablers) compared to that of Indonesian and Vietnamese respondents.” and “Cost-related barriers are more common in more resource-limited countries, while many barriers are not directly related to cost.”

The following sentence has been added into the discussion section, “It is also worth noting that the quotes related to the cost-related barriers are more common in Indonesian and Vietnamese respondents than in Thai respondents.”

VERSION 2 – REVIEW

REVIEWER	Haukka, Kaisa University of Helsinki, Dept Microbiology
REVIEW RETURNED	14-Jan-2024
GENERAL COMMENTS	The authors replied to all my comments carefully and I found their detailed answers and additions to be very useful for understanding the work and its results better. In it's current form the manuscript is a fine example of using TDF to help designing relevant interventions and improving diagnostic stewardship practices.